# RayFusion: Ray Fusion Enhanced Collaborative Visual Perception

**Shaohong Wang[1], Bin Lu[1], Xinyu Xiao[2], Hanzhi Zhong[1], Bowen Pang[1],**
**Tong Wang**[1], **Zhiyu Xiang**[1], **Hangguan Shan**[1], **Eryun Liu**[1]*

[1]Zhejiang University  [2]Institute of Automation of Chinese Academy of Sciences

wangsh0111@zju.edu.cn, eryunliu@zju.edu.cn

## Abstract

Collaborative visual perception methods have gained widespread attention in the autonomous driving community in recent years due to their ability to address sensor limitation problems. However, the absence of explicit depth information often makes it difficult for camera-based perception systems, e.g., 3D object detection, to generate accurate predictions. To alleviate the ambiguity in depth estimation, we propose RayFusion, a ray-based fusion method for collaborative visual perception. Using ray occupancy information from collaborators, RayFusion reduces redundancy and false positive predictions along camera rays, enhancing the detection performance of purely camera-based collaborative perception systems. Comprehensive experiments show that our method consistently outperforms existing state-of-the-art models, substantially advancing the performance of collaborative visual perception. The code is available at https://github.com/wangsh0111/RayFusion.

## 1 Introduction

V2X-based collaborative perception, enabled by information sharing among agents, has been proven to be beneficial for enhancing perception performance in autonomous driving systems. It effectively alleviates occlusions, mitigates depth estimation ambiguity in individual agent systems, extends perception range, and improves the performance of camera-based 3D object detection [35, 5, 23, 6, 33, 22, 29]. Existing mainstream camera-based collaborative perception methods [6, 33, 22, 29] typically construct a unified and dense bird's-eye view (BEV) representation through collaborative communication (e.g. sharing BEV features or image features from individual agents) and then perform object detection in BEV space. Some query-based approaches [1, 3] achieve collaboration by conducting attention-based interactions among multi-agent query features. However, existing

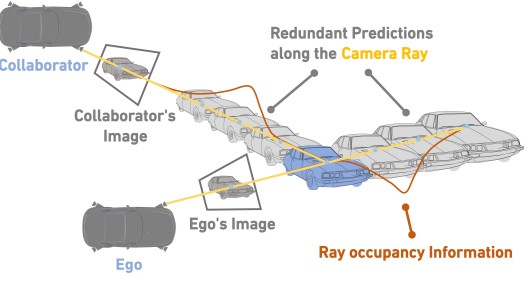

Figure 1: Due to the ambiguity in depth estimation, individual agent typically predicts multiple targets along the camera ray. However, when an instance is observed by multiple agents, they can record the target's occupancy information along the ray and cross-validate its true 3D position.

camera-based collaborative perception methods often do not explicitly model the camera imaging process of the same object from different agent viewpoints. This limitation hinders accurate cross-view

---

*Corresponding author.

39th Conference on Neural Information Processing Systems (NeurIPS 2025).

validation of an object's true 3D location, reduces the network's ability to distinguish hard negative samples along the camera rays, and ultimately leads to suboptimal perception performance.

In this work, we propose RayFusion, a ray-based fusion method for collaborative visual perception. As illustrated in Figure 1, individual agent typically tends to predict multiple targets along the camera rays. However, when an instance is observed by multiple agents, the ray occupancy information from different perspectives enables precise localization of the instance in 3D space. Here, ray occupancy information refers to whether certain points along camera rays in 3D space are occupied by an object, representing the occupancy state along the ray's trajectory. Based on this insight, our key idea is to leverage the ray occupancy information of instances to reduce redundancy and false positive predictions along camera rays, thereby enhancing the network's ability to distinguish hard negative samples and accurately localize true positive samples. Specifically, RayFusion consists of three key designs: i) Spatial-temporal alignment, which achieves spatial alignment of agent information using a unified coordinate system and improves robustness to communication delays by modeling motion. ii) Ray occupancy information encoding, obtaining an explicit representation of ray occupancy for each instance by incorporating depth information into camera rays. This explicitly accounts for the 3D structure of the scene and the potential real 3D positions of instances, which helps improve the network's depth estimation capability and consequently enhances its ability to accurately localize instances. iii) Multi-scale instance feature aggregation, which enables effective multi-agent interaction by capturing both local and global spatial features in parallel, enhancing robustness to localization noise and ultimately achieving superior collaborative perception performance.

To evaluate RayFusion, we conduct extensive experiments on a real-world dataset, DAIR-V2X [38], and two simulated datasets, V2XSet [35] and OPV2V [36]. The experimental results demonstrate that RayFusion significantly outperforms previous works in terms of both performance and robustness across multiple datasets. In summary, our contributions are as follows:

1. We introduce RayFusion, a superior camera-only collaborative perception framework based on ray occupancy information fusion. This framework enhances the network's ability to distinguish hard negative samples and accurately localize true positive samples, thereby improving the detection performance of collaborative visual perception systems.

2. We design a spatial-temporal alignment module that uses a unified coordinate system to achieve spatial alignment of agent information while modeling motion to enhance the system's robustness to communication delays.

3. We propose a ray occupancy information encoding module that explicitly accounts for the 3D structure of the scene and the potential precise 3D positions of instances. This reduces redundancy and false positive predictions while simultaneously enhancing the network's ability to localize instances along camera rays.

4. We develop a multi-scale instance feature aggregation module, enabling effective interaction with multi-agent instance features to achieve superior collaborative perception performance. Our method, RayFusion, achieves state-of-the-art performance on a real-world dataset, DAIR-V2X, and two simulation datasets, V2XSet and OPV2V, with AP70 improvements of 3.64, 3.47, and 8.21 over the previous state-of-the-art methods, respectively.

## 2 Related work

### 2.1 Collaborative perception

Collaborative perception holds significant potential for enhancing the safety of autonomous driving, with intermediate fusion strategies gaining widespread attention due to their potential to achieve optimal performance under limited and controlled communication resources. V2VNet [31] leverages a spatially aware graph neural network to aggregate shared feature representations among multiple agents. V2X-ViT [35] introduces a heterogeneous multi-agent attention module to enable adaptive information fusion between heterogeneous agents. Where2comm [5] transmits essential perceptual information via a sparse spatial confidence map to reduce communication bandwidth consumption. TransIFF [1] further reduces communication bandwidth consumption by transmitting object queries. CoAlign [23] enhances robustness in collaborative perception by correcting pose errors. CoCa3D [6] leverages multi-view information from collaborators to alleviate the depth estimation ambiguity of a single agent. However, its potential for further enhancement is limited as it does not explicitly

consider the 3D structure of the scene. IFTR [29] interacts with multi-view images from multiple agents using a predefined BEV grid, facilitating the advancement of budget-constrained collaborative systems. HM-ViT [33] and HEAL [22] explore multi-agent heterogeneous modality collaborative perception, further expanding the scale of collaboration.

However, existing collaborative visual perception methods often do not fully exploit the 3D structural information of the scene and the multi-view imaging process of the same object from different agent perspectives. As a result, they struggle to resolve depth estimation ambiguities, making it difficult for the network to distinguish true positive and false positive samples along camera rays, which ultimately leads to suboptimal perception performance. In this paper, we propose RayFusion, a novel framework that explicitly models the 3D scene structure and the camera imaging process from multiple agent viewpoints. By performing cross-view validation, RayFusion effectively distinguishes true positives and false positives, thereby achieving more efficient and practical collaborative perception.

## 2.2 Camera-based 3D object detection

Generally speaking, there are two research directions on camera-based 3D object detection, i.e., dense feature based algorithms and sparse query based algorithms. Dense algorithms represent the main research direction in camera-based 3D detection, leveraging dense feature vectors for view transformation, feature fusion, and box prediction. Currently, BEV-based approaches form the core of dense algorithms. LSS [25] and CaDDN [27] utilize view transformation modules to convert dense 2D image features into BEV space through forward projection. BEVDet [8, 7] and BEVDepth [10] employs a lift-splat operation for view transformation and further encodes BEV features using a BEV encoder. BEVFormer [11, 37] generates BEV features through deformable attention in a backward projection manner, circumventing reliance on explicit depth information. FB-BEV [12] and DualBEV [9] effectively combine the advantages of forward and backward projection methods to generate higher-quality BEV features, leading to enhanced perception performance.

Compared to dense algorithms, the sparse query-based paradigm have recently gained significant attention in the community due to their low computational complexity. DETR3D [32] is a representative sparse approach that utilizes a set of sparse 3D query vectors to sample and fuse multi-view image features. The PETR series [18, 19, 30] introduce 3D position encoding, leveraging global attention for direct multi-view feature fusion and conducting temporal optimization. SparseBEV [17] and Sparse4D [14, 15, 16] enhances DETR3D through multi-point feature sampling and temporal fusion, thereby improving perception performance.

# 3 RayFusion

As depicted in Figure 2, RayFusion comprises a single-agent detector, a collaborative message generation module, a spatial-temporal alignment module, a ray occupancy information encoding module, a multi-scale instance feature aggregation module, and a detection head. The single-agent detector and the collaborative message generation module generate instance information from multi-view input images for communication and collaboration. The spatial-temporal alignment module performs spatial-temporal alignment of instance information across agents by employing a unified coordinate system and modeling motion. The ray occupancy information encoding module leverages multi-view information from multiple agents to mitigate depth estimation ambiguity. The multi-scale instance feature aggregation module enables effective interaction among instance features through multi-scale feature fusion. Finally, the updated instance features are fed into the detection head to predict the target's category and location information.

## 3.1 Collaborative message generation

**Single-agent camera-only 3D object detector.** RayFusion adopts Sparse4D [16] as the single-agent detector, which consists of an image encoder, a depth prediction head, a decoder, and a detection head. The image encoder is a standard 2D backbone (ResNet50 [4]) used to extract semantic features from multi-view images of agent $j$. The depth prediction head estimates dense depth distribution $\mathcal{D}_j^k$ for the $k$-th camera view and is supervised using LiDAR point clouds to accelerate convergence. The decoder then initializes $N$ learnable object queries $Q_0 \in \mathbb{R}^{N \times C}$ in 3D space, which interact with multi-view image features to iteratively update instance representations. Finally, the decoder's output

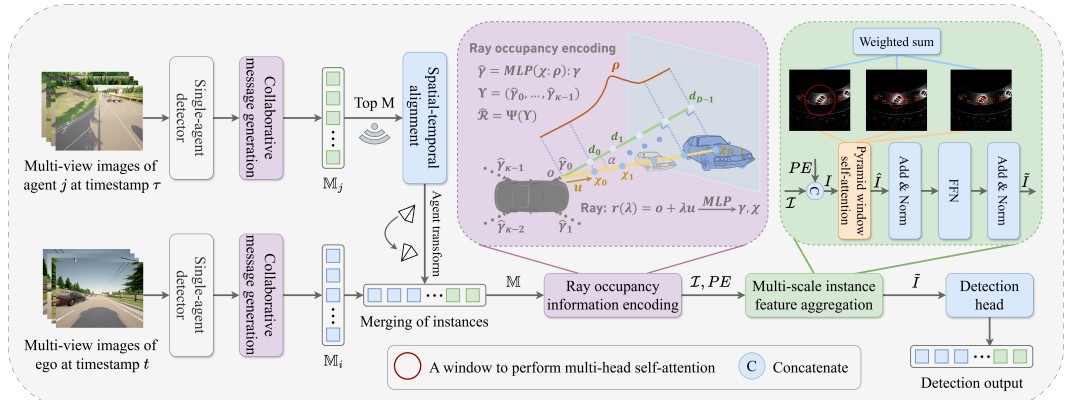

Figure 2: Overall architecture of RayFusion. i) The single-agent detector and the collaborative message generation module generate instance information for communication and collaboration; ii) The spatial-temporal alignment module enhances system robustness to latency by modeling motion; iii) The ray occupancy information encoding module leverages multi-view information to mitigate depth estimation ambiguity; iv) The multi-scale instance feature aggregation facilitates effective interaction among instance features, promoting comprehensive and precise collaborative perception.

instance features $\mathcal{I}_j$ are processed by a detection head (a multi-layer perceptron, MLP) to predict the target's category and anchor predictions $\mathcal{A}_j = [x, y, z, \ln w, \ln h, \ln l, \sin yam, \cos yam, v_x, v_y, v_z]$.

**Collaborative message generation.** Given the outputs of the single-agent detector (i.e., $\mathcal{D}_j, \mathcal{I}_j, \mathcal{A}_j$), the collaborative message shared via communication is $\mathbb{M}_j = \{\mathcal{I}_j, \mathcal{A}_j, \mathcal{R}_j\}$, where $\mathcal{R}_j$ denotes the ray occupancy information. Specifically, $\mathcal{R}_j = \{o_j^k, u_j^k, \alpha_j^k, \rho_j^k \mid k = 0, 1, \ldots, \kappa\}$, where $o_j^k$ denotes the coordinate of the camera origin, $u_j^k$ represents the direction vector of the camera ray (i.e., the ray from the camera origin to the center of the instance anchor), $\alpha_j^k$ is the angle between the camera ray and the optical axis, $\rho_j^k$ indicates the occupancy state along the camera ray, and $\kappa$ is the number of surround-view cameras. To obtain $\rho_j^k$, we project the center of the instance anchor onto the corresponding depth map $\mathcal{D}_j^k$, and apply bilinear sampling. This process is formulated as:

$$\rho_j^k = \text{Bilinear}(\mathcal{D}_j^k, \mathcal{P}_j^k(x, y, z)) \in \mathbb{R}^D \tag{1}$$

where $\mathcal{P}_j^k(x, y, z)$ represents the pixel coordinates of the instance anchor center projected onto the $k$-th image view, $D$ denotes the number of discrete depth bins. During the collaborative perception process, agent $j$ selects the top $M$ instances with the highest confidence scores and broadcasts their corresponding instance information for communication and collaboration.

### 3.2 Ray-driven position encoding

Position encoding is crucial for instance-level fusion, as it introduces location information during feature fusion to compensate for the limitations of the attention mechanisms [28]. However, previous works typically adopt learnable or sinusoidal position encoding, which are inadequate for effectively modeling the camera imaging process of instances in 3D space. In our proposed RayFusion, the position encoding consists of two components: vanilla position encoding and ray occupancy information encoding. The vanilla position encoding maps the anchor into a high-dimensional space using a lightweight anchor encoder $\Phi$ (MLP). Meanwhile, the ray occupancy information encoding explicitly represents the ray occupancy state of the corresponding instance, as detailed in Section 3.2.2.

#### 3.2.1 Spatial-temporal alignment and merging of instances

**Spatial-temporal alignment.** After receiving the shared message $\mathbb{M}_j$ from agent $j$, the first step is to align instance information with respect to ego. Since different agents represent the 3D space using distinct coordinate systems, discrepancies naturally arise in the structured information descriptions (i.e., $\mathcal{A}_j, \mathcal{R}_j$) of the same instance across agents. Therefore, we choose the ego coordinate system as the unified coordinate system. Moreover, given that instance image features $\mathcal{I}_j$ are decoupled from

their structured information, we only need to align the structured information, leaving the instance image features $\mathcal{I}_j$ intact to achieve spatial-temporal alignment across agents. Thus, our key idea is to handle the structured information of instances by unifying the coordinate systems of all agents and modeling motion, thereby achieving spatial-temporal alignment among agents, facilitating effective multi-agent collaboration and enhancing the system's robustness to communication delays.

In autonomous driving, there are two types of motion: instance motion and ego motion. Instance motion refers to the movement of other instance in the environment around the ego, while ego motion refers to the motion of the ego itself within the environment. For simplicity, we model the instance motion as uniform over short periods. Given the received anchor $\mathcal{A}_j$, we first use the instance's velocity attribute to warp its position to the current frame, thereby compensating for instance motion:

$$[x, y, z]_t = [x, y, z]_\tau + [v_x, v_y, v_z]_\tau \cdot (t - \tau) \tag{2}$$

where $t$ and $\tau$ represent the timestamp of the ego and the collaborator, respectively.

Next, we warp the position, velocity, and rotation attributes of collaborator instances to the unified coordinate system in the current frame, based on the ego pose changes. This compensates for ego motion and achieves spatial alignment:

$$[x, y, z] = R_{(\tau \to t)}[x, y, z]_t + T_{(\tau \to t)} \tag{3}$$

$$[v_x, v_y, v_z] = R_{(\tau \to t)}[v_x, v_y, v_z]_\tau + T_{(\tau \to t)} \tag{4}$$

$$[\cos yaw, \sin yaw, 0] = R_{(\tau \to t)}[\cos yaw, \sin yaw, 0]_\tau \tag{5}$$

where $R_{(\tau \to t)}$ and $T_{(\tau \to t)}$ are the rotation and translation matrices from the collaborator's frame to the unified coordinate system in the current frame. When there is no communication delay (i.e., $t = \tau$), the above operations reduce to a coordinate transformation from the collaborator's frame into the unified coordinate system, achieving spatial alignment of instance anchors. Notably, the alignment of ray occupancy information differs from that of anchors. Specifically, only the camera center $o$ and the camera ray direction vector $u$ are transformed into the unified coordinate system, without performing motion compensation, as this would disrupt the original scene imaging process.

**Merging of instances.** We concatenate the aligned instance information from collaborators with all instances extracted by the ego along the sequence dimension, resulting in the merged instance information $\mathbb{M} = \{\mathcal{I}, \mathcal{A}, \mathcal{R}\}$, where $\mathcal{I}$, $\mathcal{A}$, and $\mathcal{R}$ denote the aligned instance features, anchors, and ray occupancy information, respectively.

### 3.2.2 Ray occupancy information encoding

Given the aligned ray occupancy information $\mathcal{R} = \{o_k, u_k, \alpha_k, \rho_k \mid k = 0, 1, \ldots, \kappa\}$, we propose the ray occupancy information encoding module to explicitly incorporate the camera imaging process of each instance from multiple agent perspectives. Camera imaging is based on light ray projection, where each ray traveling through 3D space may be blocked by objects or pass through different media. As shown in Figure 1, a single agent perceives the object from a single viewpoint, typically allowing it estimate the object's presence along the direction of the light ray but not its precise depth position along the ray. When multiple agents observe the same object, each camera records the occupancy information of the object along the direction of its respective light rays. Since the ray directions from different cameras vary, cross-verification among multiple agents enables accurate 3D localization of the object by identifying the intersection of multiple rays. This improves the network's ability to distinguish hard negative samples and improves its localization accuracy for true positive samples (see Figure 4). Based on this observation, we propose a ray occupancy information encoding module, as illustrated in Figure 2. This mechanism consists of three components: ray encoding, occupancy information encoding, and multi-camera ray occupancy information fusion. For simplicity, we omit the camera index in the ray encoding and occupancy information encoding.

**Ray encoding.** A camera ray can be uniquely represented by the camera's optical center coordinate $o \in \mathbb{R}^3$ and the direction vector of the ray $u \in \mathbb{R}^3$, formulated as $r(\lambda) = o + \lambda \cdot u(\lambda \geq 0)$. However, as noted in [24, 26], neural networks tend to favor low-frequency function learning. Directly learning the optical center coordinates and direction vectors to represent the ray would lead to suboptimal performance in capturing high-frequency variations. Therefore, applying a high-frequency function to map the input into a high-dimensional space before feeding it into the network can improve data

fitting. Here, we first define a mapping function $f_L$ to transform the real number $a$ from $\mathbb{R}$ to a higher-dimensional embedding space $\mathbb{R}^{2L+1}$, defined as follows:

$$f_L(a) = (a, \sin(2^0\pi a), \cos(2^0\pi a), \ldots, \sin(2^{L-1}\pi a), \cos(2^{L-1}\pi a)) \in \mathbb{R}^{2L+1} \qquad (6)$$

The function $f_L(\cdot)$ is applied separately to each of the three element in both the camera's optical center $o$ and the ray's direction vector $u$. The resulting embeddings are concatenated along the channel dimension and then processed through an MLP to obtain the ray encoding for the given instance. This process can be formalized as follows:

$$\gamma = \text{MLP}(f_L(o) : f_L(u)) \in \mathbb{R}^C \qquad (7)$$

where : denotes concatenation along the channel dimension. In our experiments, we set $L = 10$ for $f_L(o)$ and $L = 4$ for $f_L(u)$.

**Occupancy information encoding.** Encoding only the camera rays corresponding to instances cannot fully represent the potential distribution of instances along the rays. Therefore, we incorporate the classification depth distribution predicted by the depth prediction head to reveal the occupancy state $\rho$ along the camera ray. First, we project the predefined uniformly distributed discrete depth bins onto the ray direction as follows:

$$\chi = \text{MLP}\left((d_0, d_1, \ldots, d_{D-1})/\cos\alpha\right) \in \mathbb{R}^D \qquad (8)$$

where $d_i$ represents the real depth corresponding to the $i$-th depth bin in the depth prediction. Next, we use an MLP to obtain the single camera ray occupancy encoding, formalized as follows:

$$\hat{\gamma} = \text{MLP}(\chi : \rho) : \gamma \in \mathbb{R}^{2C} \qquad (9)$$

**Multi-camera ray occupancy information fusion.** When an agent has $\kappa$ camera inputs, the same instance receives different ray occupancy encodings from each camera, denoted as $\Upsilon = (\hat{\gamma}_0, \hat{\gamma}_1, \ldots, \hat{\gamma}_{\kappa-1}) \in \mathbb{R}^{\kappa \times 2C}$. However, since an instance may only appear in the field of view of a subset of cameras, we adopt an attention-based network $\Psi$ to fuse the multi-camera ray occupancy encodings into a unified representation. This process can be formalized as follows:

$$W = \text{Softmax}\left(Mean\left(\frac{Q \cdot K^T}{\sqrt{C}}, dim = -1\right) + \mathcal{M}\right) \in \mathbb{R}^{\kappa} \qquad (10)$$

$$\hat{\mathcal{R}} = MLP(w_0\hat{\gamma}_0 + w_1\hat{\gamma}_1 + \cdots + w_{\kappa-1}\hat{\gamma}_{\kappa-1}) \in \mathbb{R}^C \qquad (11)$$

where $Q, K \in \mathbb{R}^{\kappa \times C}$ are high-dimensional vectors obtained by applying projection matrices to $\Upsilon$, $w_i$ represents the $i$-th component of $W$. The mask matrix $\mathcal{M} \in \mathbb{R}^{\kappa}$ indicates whether an instance is in the field of view of a camera, where 0 indicates the instance is within the camera's field of view, and $-\infty$ indicates it is outside. Notably, in order to account for communication delays without disrupting the original scene imaging process, we follow the approach of [35] to incorporate the delay information by encoding it with sinusoidal embeddings $p_{t-\tau} \in \mathbb{R}^C$. In summary, the ray-driven positional encoding in RayFusion can be formalized as:

$$PE = \Phi(\mathcal{A}) + \hat{\mathcal{R}} + p_{t-\tau} \in \mathbb{R}^C \qquad (12)$$

### 3.3 Multi-scale instance feature aggregation

Given the instance features $\mathcal{I}$ and the corresponding ray-driven positional encodings $PE$, we first concatenate them along the channel dimension to form the input $I \in \mathbb{R}^{(N+LM)\times 2C}$ for the multi-scale instance feature aggregation module, where $L$ denotes the number of collaborators. After fusing the ray occupancy information, the instance features explicitly incorporate the 3D structure of the scene and the potential actual 3D position of the instance. Through effective interaction of instance features, the ambiguity in instance depth estimation can be significantly alleviated, reducing redundancy and erroneous detections along the camera rays. As shown in Figure 2, to achieve efficient interaction between multi-agent instance features, we propose pyramid window self-attention, which enhances the robustness of the system against localization noise by enabling multi-scale interactions between multi-agent instance features, thereby capturing both local and global spatial features in parallel. The process of $b$-th branch is formulated as follows:

$$\hat{I}_b = \text{Softmax}\left(\frac{Q \cdot K^T}{\sqrt{d}} + g(D_{(i,j)} < r_b)\right) \cdot V \in \mathbb{R}^{(N+LM)\times C} \qquad (13)$$

Table 1: 3D detection performance comparison on the DAIR-V2X [38], V2XSet [35], and OPV2V [36] datasets under perfect settings. CoSparse4D aligns multi-agent instances (see Section 3.2.1) and applies a detection head (MLP) to produce the perception results. † indicates using Sparse4D [16] as the single-agent detector.

| Method | DAIR-V2X | | V2XSet | | OPV2V | |
|---|---|---|---|---|---|---|
| | AP50 | AP70 | AP50 | AP70 | AP50 | AP70 |
| No Collaboration | 10.74 | 1.64 | 30.37 | 13.79 | 45.94 | 25.56 |
| Late Fusion | 18.57 | 5.16 | 51.41 | 25.59 | 77.62 | 51.92 |
| V2VNet (ECCV'20) | 15.26 | 2.97 | 59.54 | 39.00 | 79.06 | 57.59 |
| V2X-ViT (ECCV'22) | 15.84 | 3.07 | 59.14 | 41.23 | 78.41 | 58.38 |
| Where2comm (NeurIPS'22) | 16.03 | 3.67 | 61.69 | 43.96 | 77.14 | 58.60 |
| CoBEVT (CoRL'22) | 15.92 | 3.18 | 58.84 | 40.81 | 80.26 | 59.34 |
| CoAlign (ICRA'23) | 16.55 | 3.23 | 64.79 | 39.64 | 80.21 | 60.46 |
| IFTR (ECCV'24) | 20.51 | 7.90 | **71.73** | 49.67 | 85.56 | 66.04 |
| CoSparse4D †(Baseline) | 23.38 | 9.17 | 65.69 | 49.33 | 79.38 | 67.43 |
| RayFusion †(Ours) | **26.29** | **11.54** | 70.32 | **53.14** | **86.59** | **74.25** |

where $Q, K, V \in \mathbb{R}^{(N+LM) \times C}$ are obtained by applying an MLP to $I$, $D_{(i,j)}$ represents the distance between the positional attributes of the $i$-th query instance and the $j$-th key instance, and $r_b$ defines the receptive field threshold for the $b$-th branch. The indicator function $g(\cdot)$ ensures that only instances within the defined receptive field contribute to the attention computation. Next, the results from multiple branches are combined through a weighted summation:

$$\hat{W} = \text{Softmax}(\text{MLP}(\hat{I}_0 : \hat{I}_1 : \cdots : \hat{I}_{B-1})) \in \mathbb{R}^B \tag{14}$$

$$\hat{I} = \hat{w}_0 \hat{I}_0 + \hat{w}_1 \hat{I}_1 + \cdots + \hat{w}_{B-1} \hat{I}_{B-1} \in \mathbb{R}^{(N+LM) \times C} \tag{15}$$

where $\hat{w}_b$ represents the $b$-th component of $\hat{W}$, and $B$ denotes the number of parallel branches. The pyramid window self-attention significantly improves the robustness of instance interactions (see Figure 3): in branches with a smaller $r_b$, attention is confined to a smaller receptive field, preserving local contextual information; while in branches with a larger $r_b$, attention operates over a larger receptive field, enabling the model to capture distant visual cues and compensate for larger localization errors. Finally, the output features of pyramid window self-attention module $\hat{I}$ is followed by a feed forward network (FFN) and an Add&Norm module to get the output $\tilde{I}$ of multi-scale instance feature aggregation module.

## 3.4 Detection head

RayFusion employs an MLP-based detection head, taking instance features $\tilde{I} \in \mathbb{R}^{(N+LM) \times C}$ as input, and producing detection results $\mathcal{O} \in \mathbb{R}^{(N+LM) \times 11}$. We utilize the $\ell_1$ loss for regression and the focal loss [13] for classification. Additionally, we incorporate a cross-entropy loss to leverage LiDAR point clouds for supervising the depth estimation task, which accelerates network convergence.

## 4 Experiments

### 4.1 Experimental setup

**Datasets.** We evaluate our proposed method on three multi-agent datasets: DAIR-V2X [38], V2XSet [35], and OPV2V [36]. DAIR-V2X [38] is an open-source real-world collaborative perception dataset with images, containing 18,000 data samples with an image resolution of $1080 \times 1920$. V2XSet [35] and OPV2V [36] are simulated datasets supporting collaborative perception with 2 to 5 connected agents, co-simulated by Carla [2] and OpenCDA [34], with an image resolution of $600 \times 800$.

**Implementation details.** We implement RayFusion following Section 3 and train it for 72 epochs using the AdamW [20] optimizer. The initial learning rate is set to $1 \times 10^{-4}$ and follows a cosine annealing decay schedule. The instance feature dimension $C$ is set to 256, and the number of discrete depth bins $D$ is set to 80, while the number of ego instances $N$ and shared instances $M$ are set to

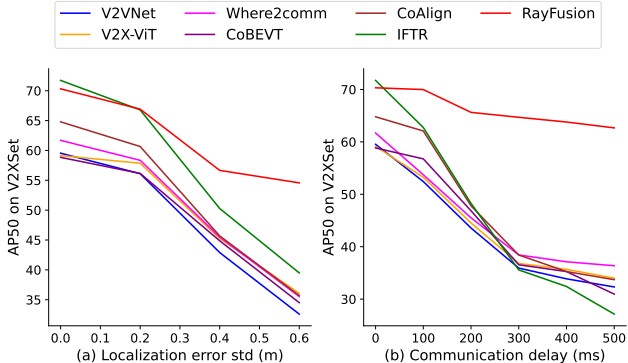

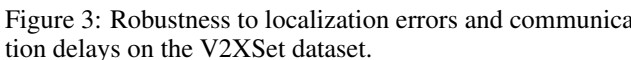

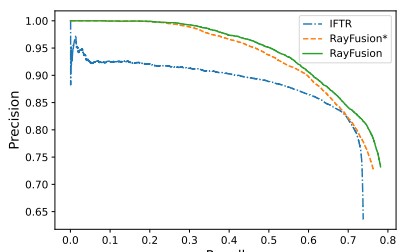

Figure 3: Robustness to localization errors and communication delays on the V2XSet dataset.

Figure 4: Precision-recall curves evaluated on the OPV2V dataset with an IoU threshold of 0.70. RayFusion* represents the removal of the ray occupancy information encoding module.

600 and 200, respectively. We utilize three pyramid windows with receptive field thresholds of 4m, 8m, and 16m. The perception range is $x, y \in [-51.2\text{m}, 51.2\text{m}]$, and the communication range is set to 70m. For BEV-based collaborative methods, we follow the implementation in IFTR [29] which adopts BEVFormer [11] as the single-agent detector. We evaluate detection performance using the Average Precision (AP) metric with Intersection over Union (IoU) thresholds of 0.50 and 0.70.

## 4.2 Main results

**Benchmark comparison.** Table 1 compares the proposed RayFusion with previous collaborative methods. We observe that the proposed RayFusion outperforms previous methods on both real-world and simulated datasets, validating the superiority of our model and its robustness to various real-world noises. Specifically, RayFusion improves upon the previous state-of-the-art method, IFTR, by 3.64, 3.47, and 8.21 in AP70 on the DAIR-V2X, V2XSet, and OPV2V datasets, respectively. Compared to previous collaborative methods, our method alleviates the ambiguity in instance depth estimation by leveraging multi-view information, enhancing the network's discriminative ability for hard negative samples along the camera rays and improving localization accuracy for true positive samples. Additionally, multi-scale instance feature aggregation facilitates effective instance interaction, promoting comprehensive and precise collaborative perception.

**Robustness to localization noise.** We follow the localization noise settings in [35, 5, 23, 29] (Gaussian noise with a mean of 0m and a variance ranging from 0m to 0.6m) and validate RayFusion's robustness to realistic localization noise in Figure 3 (a). We observe that as localization errors increase, the performance of all intermediate collaborative methods deteriorates due to the mismatch in spatial feature information. However, RayFusion significantly outperforms previous SOTAs in terms of the performance degradation slope. This is because the pyramid window self-attention in the multi-scale instance feature aggregation module captures local and global spatial features in parallel, enhancing the system's robustness to localization noise.

**Robustness to communication delays.** In autonomous driving, both instance motion and ego motion can lead to feature fusion errors under communication delays, impairing collaborative perception performance. Figure 3 (b) analyzes the detection performance of RayFusion under varying communication delays (ranging from 0 to 500ms). Notably, as communication delays increase, all other intermediate fusion methods inevitably suffer significant performance degradation due to incorrect feature matching. However, RayFusion significantly outperforms previous SOTAs under communication latency and even surpasses some fusion methods without latency (such as V2VNet and CoBEVT) under severe latency conditions (500ms). This robustness is achieved by modeling motion to handle the structured information of instances, achieving temporal alignment of instance features across agents and enhancing the system's resilience to communication delays.

**Precision-Recall Analysis.** We assess the ability of RayFusion to reduce false positive predictions along the camera rays and improve localization accuracy for true positive samples by plotting precision-recall curves. Figure 4 shows that the ray occupancy information encoding module consistently improves precision across all recall levels, and enhances recall across all precision levels

under an IoU threshold of 0.70. These results validate the effectiveness of RayFusion in reducing false positive predictions and improving the localization accuracy for true positive samples.

## 4.3 Ablation studies

**Effectiveness of different modules in RayFusion.** Here we investigate the effectiveness of various components in RayFusion, with the base model being simply merging multi-agent instance features and applying an MLP to produce 3D detection results. As shown in Table 2: i) STA achieves an AP70 improvement of 28.99 on OPV2V and 4.19 on DAIR-V2X by aligning the structured instance information from multiple agents, thereby enabling effective instance feature fusion; ii) MIFA achieves improvements of 4.61 and 1.03 in AP70 on OPV2V and DAIR-V2X, respectively, by fusing multi-scale information and enhancing the semantic understanding of instance features; iii) ROE further contributes performance gains of 2.21 and 1.34 in

Table 2: Ablation study results on the OPV2V and DAIR-V2X datasets. STA, MIFA, ROE represent: i) spatial-temporal alignment, ii) multi-scale instance feature aggregation, and iii) ray occupancy information encoding, respectively. IFA replaces the pyramid window self-attention in MIFA with vanilla multi-head self-attention.

| STA | MIFA | ROE | IFA | OPV2V | | DAIR-V2X | |
|---|---|---|---|---|---|---|---|
| | | | | AP50 | AP70 | AP50 | AP70 |
| | | | | 57.80 | 38.44 | 15.11 | 4.98 |
| ✓ | | | | 79.38 | 67.43 | 23.38 | 9.17 |
| ✓ | ✓ | | | 85.40 | 72.04 | 24.86 | 10.20 |
| ✓ | ✓ | ✓ | | **86.59** | **74.25** | 26.29 | **11.54** |
| ✓ | | ✓ | ✓ | 86.40 | 73.04 | **26.31** | 10.41 |

AP70 on OPV2V and DAIR-V2X, respectively. This is achieved by leveraging multi-view information from multiple agents to cross-verify the true 3D positions of instances, which mitigates ambiguity in depth estimation and strengthens the model's ability to localize objects along camera rays. Additionally, replacing the pyramid window self-attention (PWA) in MIFA with a vanilla multi-head self-attention mechanism leads to suboptimal results, as the pyramid window design in PWA enables parallel capture of both local and global spatial features, enriching feature representations with more comprehensive contextual information.

**Analysis of different components in ROE.** In Table 3, we analyze the effectiveness of each component in ROE. The results indicate that: i) the ray encoding based on optical center coordinates and direction vectors is crucial for the effectiveness of ROE, as it explicitly represents the imaging path of instances and provides potential true position information in 3D space; ii) occupancy information encoding further enhances the performance of ROE, as the occupancy states along the ray reflect the possible distribution of instances, offering a stronger prior for multi-view cross-validation of instance positions; iii)

Table 3: Analysis of different components in ROE on the OPV2V and DAIR-V2X. RE, OE represent: i) ray encoding and ii) occupancy information encoding, respectively. WH represents the removal of high-dimensional mapping in ray encoding.

| RE | OE | WH | OPV2V | | DAIR-V2X | |
|---|---|---|---|---|---|---|
| | | | AP50 | AP70 | AP50 | AP70 |
| | | | 85.40 | 72.04 | 24.86 | 10.20 |
| ✓ | | | 86.43 | 72.87 | 25.52 | 10.63 |
| ✓ | ✓ | | **86.59** | **74.25** | **26.29** | **11.54** |
| | ✓ | ✓ | 86.18 | 73.72 | 25.87 | 11.02 |

removing the high-dimensional mapping in ray encoding, i.e., encoding optical center coordinates and direction vectors solely through an MLP leads to performance degradation, as the high-dimensional mapping projects the high-frequency information of camera rays into a higher-dimensional space, thereby reducing the learning complexity of the network.

## 4.4 Qualitative evaluation

We conduct a qualitative analysis of the RayFusion's performance using representative samples from the V2XSet dataset, as shown in Figure 5. Overall, RayFusion produces more precise results that closely align with the ground truth, while reducing redundant predictions along the camera rays. This improvement can be attributed to: i) ROE, which effectively mitigates ambiguity in instance depth estimation, thereby enhancing localization accuracy along the camera rays; and ii) the pyramid window design in multi-scale instance feature aggregation module, which captures both local and global spatial features in parallel, leading to a more robust semantic representation of instance features. Please refer to the supplementary materials for video visualization results on the DAIR-V2X, V2XSet, and OPV2V datasets.

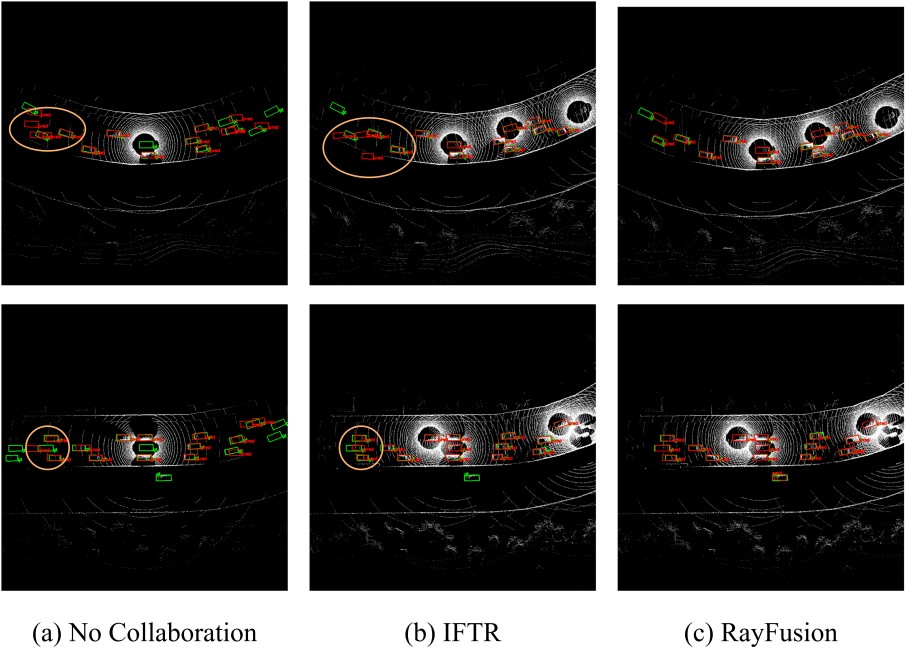

(a) No Collaboration          (b) IFTR          (c) RayFusion

Figure 5: Visualization of predictions from (a) No Collaboration, (b) IFTR and (c) RayFusion on the V2XSet test set. Green and red 3D bounding boxes represent the ground truth and prediction, respectively.

## 5  Conclusion

RayFusion is a ray-based fusion method designed to enhance the detection performance of camera-only collaborative perception systems by leveraging ray occupancy information from collaborators. It consists of three key components: spatial-temporal alignment module, ray occupancy information encoding module, and multi-scale instance feature aggregation module. The spatial-temporal alignment module achieve spatial-temporal alignment across agents, improving the system's robustness to communication delays. The ray occupancy information encoding module mitigates depth estimation ambiguities through multi-view information, enhancing the network's ability to distinguish hard negative samples along camera rays and improving the localization of true positive samples. The multi-scale instance feature aggregation module captures both local and global spatial features in parallel using pyramid windows, enabling effective instance feature interaction. Extensive experiments demonstrate the superiority of RayFusion and the effectiveness of its key components.

## Limitation and future work

In this work, we utilize the pose information of each agent to compute transformation matrices for effective multi-agent information interaction. In future work, we will explore collaborative perception under unknown poses to better protect the privacy of participating agents. Additionally, We plan to improve the system's robustness to communication delays by developing more effective occupancy representations — for instance, predicting future occupancy representations by propagating historical information through feature flow.

## Acknowledgements

This work is supported in part by Zhejiang Province Key R&D programs, China (Grant No. 2025C01039, 2024C01017, 2024C01010), in part by the National Natural Science Foundation of China under Grants U21B2029, and in part by Zhejiang Provincial Natural Science Foundation of China under Grant LR23F010006.

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

# A The efficiency of RayFusion

In RayFusion, we employ Sparse4D as the single-agent detector, which projects sparse 3D reference points onto the image plane for feature sampling. This design eliminates the need for complex view transformations and dense feature extraction, significantly reducing computational overhead. Table 4 compares RayFusion with the previous state-of-the-art method IFTR on the OPV2V dataset in terms of detection performance and inference speed. IFTR builds a dense and unified BEV representation through communication and then performs object detec-

Table 4: 3D detection performance and speed comparison on the OPV2V dataset, reporting AP metrics by default under perfect settings. The image input size for both methods is set to 640 × 480, and FPS is measured on a single GeForce RTX 3090 using the PyTorch fp32 backend.

| Method | AP50 | AP70 | FPS |
|---|---|---|---|
| IFTR | 85.56 | 66.04 | 3.36 |
| RayFusion | **86.59** | **74.25** | **8.79** |

tion in the BEV space using BEVFormer as the single-agent detector. As shown in the Table 4, RayFusion achieves superior performance with faster inference speed. These gains are attributed to the adoption of a sparse-paradigm single-agent detector and a streamlined multi-agent information fusion strategy. We advocate for the broader use of sparse-paradigms detector to achieve a better balance between performance and efficiency, facilitating the deployment of collaborative perception systems on resource-constrained edge devices.

# B Trade-off between detection performance and communication cost

We evaluate communication cost using the number of bytes in the shared messages. Under the OPV2V experimental settings in Table 1, a single agent communication cost per frame for RayFusion, IFTR, and late fusion are 276.6 KB, 18.8 MB, and 2.7 KB, respectively. Other intermediate fusion methods incur a communication cost of approximately 64.0 MB per frame. Benefiting from interpretable instance-level collaboration, RayFusion can adapt to different bandwidth conditions by adjusting the number of shared instances $M$ among collaborators. As shown in 5, we explore the trade-off between detection per-

Table 5: The trade-off between detection performance and communication cost of RayFusion on the OPV2V datasets. $M$ is the number of shared instances among collaborators.

| $M$ | AP50 | AP70 | Bytes |
|---|---|---|---|
| 200 | **86.59** | **74.25** | 276.6KB |
| 100 | 86.50 | 74.17 | 138.3KB |
| 50 | 86.47 | 73.85 | 69.1KB |
| 25 | 86.34 | 73.74 | 34.6KB |
| 10 | 84.66 | 72.00 | 13.8KB |
| 5 | 77.73 | 63.87 | **6.9KB** |

formance and communication bandwidth of RayFusion on the OPV2V datasets. The results indicate that RayFusion achieves excellent trade-off between detection performance and communication cost. When the number of shared instances among collaborators $M$ is set to 5, it requires only 6.9 KB of communication data to enable collaborative perception.

# C The reasonableness of modeling instance motion as uniform

Following the default settings in Section 4 ($M = 200$), each agent transmits 276.6 KB of information per frame. A commonly used V2X communication technology, IEEE 802.11P-based DSRC, provides a transmission bandwidth of 27 Mbps [21]. Assuming five agents are collaborating, each agent is allocated an average bandwidth of 5.4 Mbps, resulting in a transmission delay of 400.2 ms for the data. As shown in Figure 3, the performance degradation of RayFusion under this latency is not significant, which demonstrates the feasibility of modeling instance motion as uniform motion. Moreover, the transmission delay can be reduced by decreasing the number of shared instances (e.g., 20.0 ms when transmitting 10 instances per frame, i.e., $M = 10$), making it more feasible to model instance motion as uniform motion under such settings.

# D Occupancy representation without additional supervision

The purpose of the ray occupancy information encoding module is to alleviate the ambiguity in instance-level depth estimation, thereby enhancing the network's ability to distinguish hard nega-

tive samples along the camera ray direction. A natural form of supervision is instance-level depth supervision, which can be applied to guide the ray occupancy information encoding module. Specifically, after incorporating the ray occupancy information into the instance features, the output $\tilde{I}$ of multi-scale instance feature aggregation module can be passed through a depth estimation head to predict the depth distribution of the instance, which can then be supervised for instances that are successfully matched with ground truth targets. However, when using a regression-based loss (e.g., $\ell_1$ loss) for depth supervision, this supervision becomes essentially a sub-task of the 3D object detection supervision task. Therefore, we do not apply explicit additional supervision for the occupancy representation.

## E    Challenges in real-world datasets

As shown in Table 1, the performance on the real-world DAIR-V2X dataset is significantly lower than on the simulated V2XSet and OPV2V datasets. This performance gap is primarily attributed to three factors: i) DAIR-V2X includes at most two collaborative agents, whereas V2XSet and OPV2V support up to five; ii) DAIR-V2X provides only front-view camera images, while V2XSet and OPV2V offer four surround-view camera inputs; and iii) real-world scenes are generally more complex than simulated ones, making depth estimation substantially more challenging.

## F    Broader impacts

- The positive impact of this work is that RayFusion leverages V2X communication technology to share instance information, alleviating the ambiguity in depth estimation, reducing redundancy and false positive predictions along camera rays. This improves the perceptual capabilities of budget-constrained vehicles regarding their surrounding environment, thereby reducing the occurrence of traffic accidents.

- In addition, RayFusion achieves spatial alignment of multi-agent instance information through a unified coordinate system and improves robustness to communication delays by modeling motion. Its multi-scale feature interaction design further enhances resilience to localization noise.

- RayFusion also reduces computational cost by adopting a sparse-paradigms single-agent detector and reduces communication overhead through instance-level information sharing, making it well-suited for real-world deployment.

- Moreover, RayFusion is inherently scalable, supporting the integration of additional agents for collaboration without requiring retraining.

- In summary, this work promotes the practical application of camera-only collaborative perception systems in autonomous driving.

