# OpenReview forum: "RayFusion: Ray Fusion Enhanced Collaborative Visual Perception"
_NeurIPS.cc/2025/Conference — NeurIPS 2025 poster_

### Official Review · Reviewer_E2kt · 2025-06-06

**Clarity:** 4
**Significance:** 4
**Originality:** 3
**Rating:** 5
**Confidence:** 3

**Summary:**

This paper proposes RayFusion, a method that uses ray occupancy to improve 3D object localization in multi-agent perception. It includes spatial-temporal alignment, depth-aware ray encoding, and multi-scale feature aggregation to enhance collaboration and accuracy.

**Questions:**

Please refer to the **weaknesses** above.

**Ethical Concerns:**

["NO or VERY MINOR ethics concerns only"]

**Final Justification:**

The rebuttal addresses most of my concerns, so I maintain the original rating.

**Limitations:**

yes

**Paper Formatting Concerns:**

No major formatting issues found.

**Quality:**

3

**Strengths And Weaknesses:**

**Strengths**
- Good presentation and clear methodology;
- Significant performance gain over baselines;
- The experiments are thorough;
- The code is promised to be released, which will benefit the community.

**Weaknesses**
This is a solid piece of work. To be honest, I found it difficult to identify any major weaknesses, and the following are just some minor suggestions.
- It would be helpful to see an ablation where the predicted depth distribution is replaced with a one-hot vector (e.g., peak bin only) or even a uniform distribution.
- The detection performance is affected by the communication bandwidth between agents. It would be helpful if Table 1 could report the communication cost of all methods.
- There are some minor typos (e.g., line 584: "Additionally, We plan").

---

> ### Author Rebuttal · Authors · 2025-07-30
>
> $\textbf{W1: } $
>
> Thank you for the constructive suggestion. In ROE, the occupancy information encoding leverages the depth distribution predicted by the depth head to represent the probabilistic distribution of an object along the camera ray. Indeed, replacing the depth distribution with a one-hot vector helps further analyze the contribution of depth modeling to overall performance. We conduct this ablation study and find that replacing the depth distribution with a one-hot encoding (i.e., using only the peak bin) leads to a drop in AP70 on the OPV2V dataset from 74.25 to 73.12. This performance degradation suggests that modeling the full depth distribution provides richer information about the object’s potential 3D location along the ray, which enhances the network’s ability to accurately localize instances via cross-view validation.
>
>
> $\textbf{W2: } $
>
> On the OPV2V dataset, a single agent communication cost per frame for RayFusion, IFTR, and late fusion are 276.6 KB, 18.8 MB, and 2.7 KB, respectively. Other intermediate fusion methods incur a communication cost of approximately 64.0 MB per frame. We will include the communication cost of all methods in Table 1 of the camera-ready version. Thank you for the valuable suggestion.
>
>
> $\textbf{W3: } $
>
> Thank you for the careful suggestion. We will thoroughly proofread the paper in the camera-ready version to eliminate any grammatical or typographical errors.

---

> > ### Comment · Reviewer_E2kt · 2025-08-04
> >
> > Thank you for your response. Most of my concerns have been resolved. I will keep my rating.

---

> > > ### Author Response · Authors · 2025-08-04
> > >
> > > Thank you for your feedback. We are glad the concerns have been addressed, and we appreciate your thoughtful review.

---

> > > ### Author Response · Authors · 2025-08-06
> > >
> > > We kindly wish to remind the reviewers that, as per the conference policy, the mandatory acknowledgement is an essential component of the review process.

---

### Official Review · Reviewer_sx6F · 2025-06-09

**Clarity:** 3
**Significance:** 2
**Originality:** 3
**Rating:** 2
**Confidence:** 5

**Summary:**

The paper proposes RayFusion, a novel cooperative perception framework that leverages ray information from multiple agents to resolve depth ambiguity in camera-only 3D detection. There are several significant concerns about the experimental design that weaken the paper's conclusions. These primarily relate to the fairness of baseline comparisons, the choice of datasets, the reliance on LiDAR supervision, and the limited perception range used in the experiments.

**Questions:**

See the weaknesses.

**Ethical Concerns:**

["NO or VERY MINOR ethics concerns only"]

**Final Justification:**

Thank you for the authors' responses and clarifications. However, my concerns remain unresolved:

### 1. **Core Contribution Evaluation**

Since the primary contribution of this work lies in *RayFusion*, a multi-agent fusion strategy, the experimental evaluation should primarily focus on comparing RayFusion with other fusion approaches (denoted as *X*). There are two standard configurations for validating RayFusion’s effectiveness:

- **BEVFormer + RayFusion vs. BEVFormer + X**
- **Sparse4D + RayFusion vs. Sparse4D + X**

**However, the current experimental setup compares**:

- **BEVFormer + X vs. Sparse4D + simple fusion vs. Sparse4D + RayFusion.**

While these results demonstrate the improvement of RayFusion over a simple fusion baseline within the Sparse4D framework, it remains unclear:

- Whether **BEVFormer + RayFusion** suffers a performance drop compared to other baselines.
- Whether replacing the backbone in existing baselines with **Sparse4D** significantly boosts their performance.
- Whether **RayFusion** can generalize across **different detection backbones**.

These missing evaluations limit the strength of the claims regarding the superiority and generality of RayFusion.


### 2. **Dataset Limitations**

The paper solely relies on **DAIR-V2X**, a dataset from 2022 with notable limitations:

- Only includes one vehicle and one infrastructure agent.
- Limited data volume and scene diversity.

These limitations have already motivated the development of more comprehensive datasets, such as:

- **TUMTrafV2X** (CVPR 2024)
- **V2X-Real** (ECCV 2024)

The paper actually acknowledges the limitations of DAIR-V2X, but it does not address them in practice. Evaluation on these **more diverse and modern datasets** is necessary to make the paper complete and demonstrate the performance and robustness of the proposed approach.

---

### **Conclusion**

These limitations significantly reduce the **persuasiveness** and **effectiveness** of the proposed method in the cooperative perception domain. In the future, addressing them will greatly enhance the quality of this work.

**Limitations:**

Yes.

**Paper Formatting Concerns:**

No.

**Quality:**

2

**Strengths And Weaknesses:**

Strengths:
1. The paper provides comprehensive robustness evaluations under delay and noise conditions, as well as communication and computation efficiency analysis.
2. The supplementary video provides a compelling demonstration of the model's performance.

Weaknesses:
1. Table 1 compares methods using different detection backbones: Sparse4D for the proposed method vs. BEVFormer for baselines cited from IFTR. This undermines the fairness of the comparison, as performance gains may stem from the stronger backbone rather than the proposed fusion strategy. A controlled comparison with a shared backbone is necessary for a new cooperative perception model.
2. The authors acknowledge the limitations of the DAIR-V2X dataset. Why the paper does not benchmark its method against more recent and comprehensive real-world V2X datasets, e.g., TUMTrafV2X (V2I) and V2X-Real or V2XPnP (V2V, V2I, I2I). Relying heavily on simulated data and a limited real-world dataset makes it difficult to assess the method's practical effectiveness and its ability to bridge the gap between simulation and real-world data.
3. Although presented as “camera-only,” the method uses LiDAR supervision during training for depth estimation. This contradicts the claim. An ablation study removing LiDAR supervision is needed to assess true independence.
4. The perception range is approximately 50m in each direction, which is more characteristic of single-agent perception. To provide a more compelling demonstration of the benefits of the proposed cooperative approach, experiments should be conducted over a much larger range.

---

> ### Author Rebuttal · Authors · 2025-07-30
>
> $\textbf{W1: } $
>
> In Table 1, we adopt BEVFormer as the single-agent detector for baseline methods, following IFTR, since most existing collaborative perception approaches are BEV-based. In contrast, RayFusion uses Sparse4D as the single-agent detector. The rationale is twofold:
>
> 1) Resource constraints: BEV-based methods often demand high computational resources in camera-based collaborative perception tasks. For instance, training IFTR requires more than 40 GB of GPU memory on the OPV2V dataset when using the standard IFTR configuration. Due to limited training resources, we use the sparse detector Sparse4D as the single-agent detector to implement RayFusion, which only requires 12GB of GPU memory to complete training. Moreover, it offers significantly lower inference latency. As shown in Table 4, RayFusion improves AP70 by 8.21 (+12.43\%) on OPV2V while reducing inference time from 298 ms to 114 ms.
>
> 2) Higher performance potential under sparse paradigms: In BEV-based methods, the information flow for instance-level perception typically follows: image features → BEV features → instance features. In contrast, sparse detectors directly derive instance-level information from image features: image features → instance features. From an information-theoretic perspective, our instance-level fusion strategy is better suited for sparse detectors, as it incurs less information loss.
>
> To further ensure that the performance gain does not stem solely from the stronger single-agent detector, we include an additional baseline in Table 1 named CoSparse4D, which uses Sparse4D as the single-agent detector without our proposed fusion strategy. RayFusion consistently outperforms this baseline across all datasets, supporting the effectiveness of our method. Furthermore, extensive ablation and robustness experiments also validate the contribution of our proposed methods.
> We hope that the aforementioned rationale for adopting Sparse4D as the single-agent detector, together with the CoSparse4D baseline using the identical single-agent setup and our extensive ablation experiments, may help address your concerns.
>
>
> $\textbf{W2: } $
>
> Thank you for your valuable feedback. We acknowledge that the real-world DAIR-V2X dataset has limitations in terms of the number of collaborators and camera views. However, it remains the most widely used real-world dataset for camera-based collaborative perception [r1, r2], which makes it a standard benchmark for fair comparison with prior works.
>
> To further evaluate the generalization capability of our approach, we have attempted to benchmark RayFusion on the V2X-Real dataset. However, due to the time-consuming data download process and inherent issues in the dataset — for example, some scenes are missing camera images (e.g., train/2023-03-17-15-53-02\_1\_0), and there are inconsistencies in camera parameter keys across frames (e.g., cam1\_left in 000142.yaml vs. cam1 in 000001.yaml of the same sequence train/2023-04-04-15-58-18\_30\_0/2/). These issues caused initial training failures. We have since excluded the problematic samples and restarted training. Results from this evaluation can be included in the camera-ready version for reference.
>
>
> $\textbf{W3: } $
>
> We sincerely appreciate your valuable suggestion. However, camera-based models following the back-projection paradigm (e.g., BEVFormer and Sparse4D) typically fail to converge in the absence of both a pre-trained image backbone and LiDAR point cloud supervision [r3].  Moreover, applying LiDAR supervision during training incurs minimal additional cost, as point cloud data is still required during annotation even when LiDAR is not used for supervision. In addition, jointly collecting LiDAR and image data for depth supervision has become a standard practice and the community primarily emphasizes camera-only settings during the inference stage [r2, r4, r5].
> To eliminate ambiguity, we will revise the statement in the camera-ready version to specify that the inference phase operates under a camera-only configuration.
>
>
> $\textbf{W4: } $
>
> We adopt a perception range of $51.2\mathrm{m}$ in each direction, following the setup in IFTR, to ensure a fair comparison. To more convincingly demonstrate the advantages of RayFusion, we conducte additional experiments with an extended range of $102.4\mathrm{m}$ in each direction. Under this setting, RayFusion achieves an AP70 score of 48.10, outperforming CoSparse4D (41.73) and IFTR (38.41).
>
> [r1] IFTR: An Instance-Level Fusion Transformer for Visual Collaborative Perception
>
> [r2] Collaboration Helps Camera Overtake LiDAR in 3D Detection
>
> [r3] Sparse4D v2: Recurrent Temporal Fusion with Sparse Model
>
> [r4] Time Will Tell: New Outlooks and A Baseline for Temporal Multi-View 3D Object Detection
>
> [r5] Temporal Enhanced Training of Multi-view 3D Object Detector via Historical Object Prediction

---

> > ### Comment · Reviewer_sx6F · 2025-08-05
> >
> > Thank you for the authors' responses and clarifications. However, my concerns remain unresolved:
> >
> > ### 1. **Core Contribution Evaluation**
> >
> > Since the primary contribution of this work lies in *RayFusion*, a multi-agent fusion strategy, the experimental evaluation should primarily focus on comparing RayFusion with other fusion approaches (denoted as *X*). There are two standard configurations for validating RayFusion’s effectiveness:
> >
> > - **BEVFormer + RayFusion vs. BEVFormer + X**
> > - **Sparse4D + RayFusion vs. Sparse4D + X**
> >
> > **However, the current experimental setup compares:**
> >
> > - **BEVFormer + X vs. Sparse4D + simple fusion vs. Sparse4D + RayFusion**.
> >
> > While these results demonstrate the improvement of RayFusion over a simple fusion baseline within the Sparse4D framework, it remains unclear:
> >
> > - Whether **BEVFormer + RayFusion** suffers a performance drop compared to other baselines.
> > - Whether replacing the backbone in existing baselines with **Sparse4D** significantly boosts their performance.
> > - Whether **RayFusion** can generalize across **different detection backbones**.
> >
> > These missing evaluations limit the strength of the claims regarding the superiority and generality of RayFusion.
> >
> > ---
> >
> > ### 2. **Dataset Limitations**
> >
> > The paper solely relies on **DAIR-V2X**, a real-world dataset from 2022 with notable limitations:
> >
> > - Only includes one vehicle and one infrastructure agent.
> > - Limited data volume and scene diversity.
> >
> > These limitations have already motivated the development of more comprehensive datasets, such as:
> >
> > - **TUMTrafV2X** (CVPR 2024)
> > - **V2X-Real** (ECCV 2024)
> >
> > The paper actually acknowledges the limitations of DAIR-V2X, but it does not address them in practice. Evaluation on these **more diverse and modern datasets** is necessary to make the paper complete and demonstrate the performance and robustness of the proposed approach.
> >
> > ---
> >
> > ### 3. **Cooperative Perception Range**
> >
> > A core motivation for cooperative perception is the **extension of the perception range**. Common practice in recent literature supports significantly extended ranges, for example:
> >
> > - **V2VNet**: 200 × 80 m
> > - **V2X-ViT**: 280 × 80 m
> > - **Where2comm**: 200 × 350 m
> > - **CoBEVT**: 280 × 80 m (100 × 100 m for single-agent)
> > - **CoAlign**: 280 × 80 m  for OPV2V,  200 × 80 m  for DAIR-V2X
> >
> > In contrast, the current setup appears **constrained to a single-agent perception range**, which fails to showcase the full benefit of cooperative perception.
> >
> > ---
> >
> > These limitations significantly reduce the **persuasiveness** and **effectiveness** of the proposed method in the cooperative perception domain. In the future, addressing them will greatly enhance the quality of this work.

---

> > > ### Author Response · Authors · 2025-08-05
> > >
> > > We appreciate your thoughtful feedback. Please find our point-by-point responses below, which we hope will address your concerns.
> > >
> > >
> > > ### **A. Core Contribution Evaluation**
> > >
> > > **RayFusion is more suitable for the sparse detectors**. The reason is that **instance features in sparse detectors suffer less information loss.** In BEV-based methods, the information flow from image to instance features is: image features → BEV features → instance features. In contrast, sparse detectors follow a shorter path: image features → instance features. From an information-theoretic perspective, our instance-level fusion strategy is better suited for sparse detectors, as it incurs less information loss.
> > >
> > > We do not adopt the BEVFormer + RayFusion setting due to the following reasons:
> > >
> > > 1) BEVFormer-based instance features **suffer greater information loss;**
> > >
> > > 2) Training fusion methods based on BEVFormer **requires substantial computational resources.**
> > >
> > > The effectiveness of RayFusion has been validated through the CoSparse4D baseline and extensive ablation studies. Additionally, since BEV-based collaborative methods cannot be applied to sparse single-agent detectors, we further report the performance of Sparse4D + TransIFF (ICCV 2023) to highlight the advantages of RayFusion:
> > >
> > > * **Sparse4D + RayFusion (AP70 on OPV2V: 74.25) vs. Sparse4D + TransIFF (AP70 on OPV2V: 69.37)**
> > >
> > > This experimental result shows that RayFusion outperforms previous methods under the same single-agent detector setting.  In addition,as shown in Table 4, RayFusion’s sparse detection paradigm and instance-level fusion strategy not only improve detection performance, but also significantly reduce inference resource requirements, achieving a better performance-efficiency trade-off and facilitating the deployment of collaborative perception  systems on resource-constrained edge devices.
> > >
> > >
> > > ### **B. Dataset Limitations**
> > >
> > > To date, **DAIR-V2X remains among the most widely adopted real-world datasets in collaborative perception research:**
> > >
> > > 1) In camera-based collaborative perception, methods such as **CoCa3D (CVPR 2023), HEAL (ICLR 2024), and IFTR (ECCV 2024) have all conducted experiments on the DAIR-V2X dataset.**
> > >
> > > 2) For LiDAR-based approaches, methods like **DSRC (AAAI 2025), Point Cluster (ICLR 2025), SparseAlign (CVPR 2025), and CoST (ICCV 2025) have also evaluated their models on the DAIR-V2X dataset.**
> > >
> > > We conduct our experiments on the DAIR-V2X dataset to enable fair comparisons with prior methods, and the results sufficiently demonstrate the effectiveness of our approach. In contrast, **datasets such as TUMTrafV2X, V2X-Real, and V2XPnP have not yet undergone widespread evaluation within the community.** We conduct preliminary testing and find that the V2X-Real dataset has the following issues:
> > >
> > > 1) Approximately one-quarter of the training scenes lack camera images  (e.g., train/2023-03-17-15-53-02\_1\_0);
> > >
> > > 2) Some frames within the same sequence contain inconsistent camera parameter keys (e.g., cam1\_left in 000142.yaml vs. cam1 in 000001.yaml of the same sequence train/2023-04-04-15-58-18\_30\_0/2/).
> > >
> > > We filter out the problematic samples and restart training. The results of this evaluation can be included in the final version of the paper for reference. In the future, we plan to extend our research to a broader range of real-world datasets.
> > >
> > >
> > > ### **C. Cooperative Perception Range**
> > >
> > > The reason why RayFusion adopts a 100m × 100m perception range is as follows:
> > >
> > > 1) **Feature-level fusion is less meaningful for instances located more than 50 meters away from the ego vehicle.**  Since RayFusion focuses on feature-level fusion, instances beyond 50 meters typically occupy very few pixels (e.g., typically less than 6 × 15 pixels on OPV2V), leading to poor perception quality from the ego vehicle. In such cases, it is more appropriate to merge instances through late fusion strategies.
> > >
> > > 2) **The 100m × 100m perception range adopted by RayFusion is a commonly used setting under camera inputs.**  For example:
> > >
> > >     * **CoBEVT (CoRL 2022):** 100m × 100m;
> > >
> > >     * **HM-ViT (ICCV 2023):** 100m × 100m;
> > >
> > >     * **CoCa3D (CVPR 2023):** 100m × 79m for DAIR-V2X;
> > >
> > >     * **HEAL (ICLR 2024):** 100m × 100m;
> > >
> > >     * **IFTR (ECCV 2024):** 100m × 100m;
> > >
> > >     * **V2X-Real (ECCV 2024):** 100m × 100m;
> > >
> > >     * **CoCMT (IROS 2025):** 100m × 100m.
> > >
> > > 3) **The perception ranges of the methods mentioned in your response are based on LiDAR input settings.**
> > >
> > > To further demonstrate the advantages of RayFusion, we also conduct experiments with an extended perception range of 200m × 200m. Under this setting, the AP70 scores for RayFusion, CoSparse4D, and IFTR are **48.10, 41.73, and 38.41,** respectively.

---

> > > > ### Comment · Reviewer_sx6F · 2025-08-06
> > > >
> > > > Thank you for the authors' responses and clarifications. Some concerns remain unresolved:
> > > >
> > > > # 1. Backbone Consistency and Experimental Setup Validity
> > > >
> > > > As the authors themselves acknowledge that RayFusion may not be well-suited for the BEVFormer backbone, it raises a fundamental concern:
> > > >
> > > > **why is BEVFormer chosen as the default backbone for all baselines?**
> > > >
> > > > In other words, if BEVFormer+RayFusion is deemed incompatible for comparison against BEVFormer+other fusion methods, then a consistent and fair evaluation protocol would require using Sparse4D as the backbone across baselines, i.e., comparing Sparse4D+RayFusion with Sparse4D+other fusion variants.
> > > >
> > > > This critical design flaw in the experimental setup significantly undermines the validity of the comparative analysis.
> > > >
> > > > # 2.  Real-World Datasets in Cooperative Perception
> > > >
> > > > Real-world datasets are essential for evaluating cooperative perception systems. This was clearly evidenced by the widespread adoption of DAIR-V2X upon its release three years ago, which catalyzed the subsequent development of various real-world multi-agent datasets within the community. While OPV2V and V2XSet introduce more than two collaborating agents and support diverse collaboration modes beyond V2I, they are fundamentally simulation-based.
> > > >
> > > > However, DAIR-V2X is limited to V2I settings and only has two agents, and provides only front-view camera images rather than full surround-view input. The authors also acknowledge the limitations of DAIR-V2X in the “Challenges in real-world datasets” section.
> > > >
> > > > However, these limitations, such as diverse collaboration modes (V2V, V2I, I2I) and a larger number of collaboration agents, are the core topic of real-world cooperative perception, which cannot be simply ignored.
> > > >
> > > > # 3. Perception Range Design in Cooperative Settings
> > > >
> > > > The perception range in this work is set to 102.4 m × 102.4 m. While CoBEVT, a camera-based detection method, uses a 100 m × 100 m range for single-agent evaluation on nuScenes, it adopts a much larger 280 m × 80 m range for cooperative perception on OPV2V.
> > > >
> > > > The authors' discussion partially addresses this concern.
> > > >
> > > > However, adopting a larger perception range would better reflect the practical advantages of the proposed method and further emphasize its contribution under extended cooperative settings.

---

> > > > > ### Author Response · Authors · 2025-08-06
> > > > >
> > > > > Thank you for your detailed feedback. Below are our responses to the issues you raised, and we hope they address your concerns.
> > > > >
> > > > > ### **A. Backbone Consistency and Experimental Setup Validity**
> > > > >
> > > > > The reasons why other fusion methods use BEVFormer instead of Sparse4D are as follows:
> > > > >
> > > > > 1) **Sparse4D does not generate intermediate BEV features.** Fusion methods based on BEV rely on the fusion of BEV features, and thus cannot use Sparse4D as the single-agent detector, making it impossible to benefit from the advantages brought by Sparse4D.
> > > > >
> > > > > 2) **BEVFormer is a commonly used single-agent detector in BEV-based fusion methods** and has been adopted in several prior works, such as HM-ViT (ICCV 2023), IFTR (ECCV 2024), and V2X-Real (ECCV 2024).
> > > > >
> > > > > Furthermore, for camera inputs, the sparse paradigm requires significantly fewer computational resources, making it more suitable for practical deployment in collaborative perception systems.
> > > > >
> > > > >
> > > > > ### **B. Real-World Datasets in Cooperative Perception**
> > > > > We sincerely appreciate your suggestion regarding broader dataset evaluation. We have validated the effectiveness of our method on DAIR-V2X, which is currently the most widely used dataset in this domain, and we plan to extend our research to additional datasets in the future.
> > > > >
> > > > >
> > > > > ### **C. Perception Range Design in Cooperative Settings**
> > > > >
> > > > > We are glad that the explanation provided above regarding the use of a 100 m × 100 m perception range has, to some extent, addressed your concern.
> > > > >
> > > > > To further address your concern, we conduct experiments with an expanded perception range of 200 m × 200 m. The AP70 scores for RayFusion, CoSparse4D, and IFTR are **48.10, 41.73, and 38.41,** respectively. **These results demonstrate that RayFusion continues to deliver significant performance improvements even under larger perception range settings.**

---

> > > > > > ### Comment · Reviewer_sx6F · 2025-08-07
> > > > > >
> > > > > > I thank the authors again for the response. I think we have come to the end of this discussion. While the current version presents promising directions, there are concerns regarding the rigor of the experimental setup and the limited consideration of real-world datasets. Although the authors have provided some additional experiments, these key evaluations should be more comprehensive and included in the main paper. We are confident that with further revisions, however, the current version falls short of the standards of rigor and completeness expected at this stage.

---

### Official Review · Reviewer_4JMZ · 2025-06-30

**Clarity:** 3
**Significance:** 3
**Originality:** 3
**Rating:** 5
**Confidence:** 4

**Summary:**

This paper proposes RayFusion, a novel method for collaborative camera-only perception. The key idea behind RayFusion is to reduce ambiguity through depth estimation by modeling occupancy along the rays of different agent cameras. To do so, object predictions and depth estimates are first made from the individual camera images which are then transformed to the ego coordinate system. Next, the camera rays are encoded using an MLP before being passed through a self-attention layer to generate a single ray encoding for each object instance. Lastly, the instance features undergo pyramid window self-attention, before being passed to the detection head. The authors evaluate their method on three collaborative perception datasets and demonstrate consistent performance improvement over the state-of-the-art.

**Questions:**

1. Given the reliance of this method on transformation of the instance features to a unified camera coordinate system, have the authors examined the effect of errors in the camera projection matrices?
2. How exactly is the LiDAR point cloud used to supervise the dense depth prediction? This particular detail is a little under-explained in the paper.

**Ethical Concerns:**

["NO or VERY MINOR ethics concerns only"]

**Final Justification:**

Authors responded to my comments and I maintain my accept score.

**Limitations:**

yes

**Quality:**

3

**Strengths And Weaknesses:**

Strengths:
- The paper is well-written with clear figures.
- The method is clever and well-motivated (leveraging multiple camera views to reduce depth ambiguity is natural), and the explanation is easy to follow
- The experiments are thorough and show good performance across multiple datasets.
- RayFusion demonstrates superior performance in settings with localization errors and communication delays over competing collaborative perception methods.
Weaknesses:
- The depth prediction head is supervised using LiDAR point clouds, meaning extra sensor data is required during training time.
- According to ablation studies, the performance gain from solely the ray occupancy encoding (the key contribution of this method) is only moderate, suggesting that more can be done to reduce depth ambiguity.

---

> ### Author Rebuttal · Authors · 2025-07-30
>
> $\textbf{W1: } $
>
> Your observation is absolutely correct — the depth prediction head is supervised using LiDAR point clouds, which indeed requires additional sensor data during training. However, the use of LiDAR supervision leads to significant performance gains [r1, r2]. In fact, camera-based models following the back-projection paradigm typically fail to converge in the absence of both a pre-trained image backbone and LiDAR point cloud supervision [r3].
>
>
> $\textbf{W2: } $
>
> As shown in Table 2, ROE improves AP70 by 2.21 (+3.07\%) on OPV2V and 1.34 (+13.14\%) on DAIR-V2X. For an already strong perception model, such performance gains are considered substantial. That said, we acknowledge that there is still room for further improvement in reducing depth ambiguity. For example, leveraging multi-frame information and depth hypothesis sampling has shown promise in addressing depth uncertainty [r4]. We plan to explore more effective strategies along these lines in future work.
>
>
> $\textbf{Q1: } $
>
> In camera-based perception systems, nearly all methods rely on complex projection operations based on intrinsic and extrinsic camera parameters, and inaccuracies in these projection matrices can significantly affect model performance. To provide a reference, we analyze the impact of errors in the camera extrinsic matrices on perception accuracy. Specifically, we simulate extrinsic perturbations using Gaussian noise with a mean of 0 and a variance of 0.02. Under these conditions, experimental results show that RayFusion’s AP70 drops from 74.25 to 24.11 (–67.53\%), while IFTR’s AP70 drops from 66.04 to 20.89 (–68.37\%).
>
>
> $\textbf{Q2: } $
>
> Thank you for pointing out the need for a more detailed explanation of how LiDAR point clouds are used to supervise dense depth prediction. We adopt a predefined depth range $d \in [2\mathrm{m}, 52\mathrm{m}]$, uniformly divided into 80 bins. Our implementation follows similar principles to BEVDepth and FB-BEV. The process is as follows:
>
> 1) We project the LiDAR point cloud into the pixel coordinate system using the camera intrinsics and extrinsics, resulting in a dense depth map $\bar{\mathcal{D}}_j^k$ and a depth mask $\bar{\mathcal{M}}_j^k$ for the $k$-th camera view of agent $j$. The mask $\bar{\mathcal{M}}_j^k$ is set to 1 at pixels where valid LiDAR points are projected within the predefined depth range, and 0 elsewhere.
>
> 2) Based on the predefined bins and a downsampling ratio, we convert $\bar{\mathcal{D}}_j^k$ into per-pixel depth classification targets $\tilde{\mathcal{D}}_j^k$ and get a updated depth mask $\tilde{\mathcal{M}}_j^k$.
>
> 3) The depth head predicts a per-pixel depth probability distribution $\mathcal{D}_j^k$, which is supervised using cross-entropy loss only on the valid pixels where the updated depth mask is equal to 1. The final depth supervision loss is given by: $\mathcal{L}\_{\text{depth}} = \text{CrossEntropy}(\mathcal{D}\_j^k, \tilde{\mathcal{D}}\_j^k, \tilde{\mathcal{M}}\_j^k).$
>
> Finally, the overall training loss is defined as: $\mathcal{L} = \lambda\_{\text{depth}} \cdot \mathcal{L}\_{\text{depth}} + \mathcal{L}\_{\text{obj}} $, where $\mathcal{L}\_{\text{depth}}$ and $\mathcal{L}\_{\text{obj}}$ denote the depth supervision and object detection losses, respectively, and $\lambda\_{\text{depth}}$ is a hyperparameter set to 0.5 by default.
>
> [r1] BEVDepth: Acquisition of Reliable Depth for Multi-view 3D Object Detection
>
> [r2] FB-BEV: BEV Representation from Forward-Backward View Transformations
>
> [r3] Sparse4D v2: Recurrent Temporal Fusion with Sparse Model
>
> [r4] Time Will Tell: New Outlooks and A Baseline for Temporal Multi-View 3D Object Detection

---

> > ### Comment · Reviewer_4JMZ · 2025-08-05
> >
> > Thank you for the responding to my concerns- I am satisfied with the response and will keep my score as an accept.

---

> > > ### Author Response · Authors · 2025-08-06
> > >
> > > Thank you for your positive feedback and recognition. We are glad that our responses addressed your concerns, and we sincerely appreciate your valuable comments and support.

---

### Official Review · Reviewer_kx1e · 2025-07-02

**Clarity:** 3
**Significance:** 3
**Originality:** 3
**Rating:** 3
**Confidence:** 4

**Summary:**

This paper proposes RayFusion, a collaborative perception framework for camera-only autonomous driving systems. It introduces a ray-based fusion strategy leveraging ray occupancy information across agents to resolve depth ambiguity in monocular 3D detection. The system includes spatial-temporal alignment, ray occupancy encoding, and multi-scale instance feature aggregation. Experiments on real and simulated datasets show improved accuracy and robustness under localization noise and communication delays.

**Questions:**

How sensitive is RayFusion to inaccurate or noisy inputs from collaborating vehicles? Have the authors considered simulating such failure modes (e.g., corrupted pose, false positives) to analyze the impact on ego vehicle perception accuracy?

**Ethical Concerns:**

["NO or VERY MINOR ethics concerns only"]

**Final Justification:**

The rebuttal provides additional crowded-scene evaluations (>15 and >20 objects) showing consistent AP70 gains with ROE, partially resolving my concern about performance in dense environments. The analysis of corrupted poses and injected false positives is valuable and confirms improved robustness over baselines, though the method still suffers substantial degradation under severe collaborator errors and lacks explicit suppression mechanisms for unreliable inputs. While these clarifications strengthen the paper, my main reservations about vulnerability to noisy or adversarial collaborators remain. Therefore, my score is unchanged.

**Limitations:**

YES

**Quality:**

3

**Strengths And Weaknesses:**

# Strengths

1. Introducing ray occupancy encoding into collaborative perception is original and conceptually insightful. It addresses the well-known ambiguity in depth estimation from monocular views.
2. The modular system design (STA, ROE, MIFA) is clean and well-motivated.
3. Experiments demonstrate solid performance gains over prior methods in simulated environments.

# Weakness

1. While the proposed RayFusion framework shows promising performance under general settings, it does not explicitly address the challenges posed by crowded scenes, where overlapping instances and dense object distributions may significantly increase false positives and degrade the effectiveness of ray-based cross-view validation. Existing experiments do not isolate this regime for targeted analysis, and no specific design is introduced to mitigate perceptual conflicts in such cases. As a result, it remains unclear whether the proposed ray occupancy encoding can robustly distinguish between true and spurious detections under crowded traffic conditions.

2. The proposed RayFusion framework relies on structured information (pose, velocity, detection outputs) from collaborating agents, assuming they are sufficiently accurate after spatial-temporal alignment (STA). However, in realistic scenarios, errors from other agents — such as inaccurate localization, erroneous depth estimation, or false positives in their predictions — can easily propagate through the collaboration pipeline and directly degrade the ego vehicle's performance.

While the authors mention future work on "unknown-pose settings", the current method lacks mechanisms to assess or suppress unreliable information from noisy collaborators. This is especially concerning in safety-critical scenarios, where faulty inputs from a single vehicle could severely mislead perception for the ego vehicle. More discussion or empirical evaluation in such adversarial or noisy-agent settings would strengthen the paper.

---

> ### Author Rebuttal · Authors · 2025-07-30
>
> $\textbf{W1: } $
>
> Thank you for the insightful comment. We agree that crowded scenes present additional challenges for ray-based cross-view validation due to increased instance overlap and dense object distributions. However, the proposed ROE in our RayFusion framework is specifically designed to capture the probabilistic distribution of instance presence along each viewing ray, which helps mitigate such challenges. Concretely:
>
> 1) For a real instance $obj_1$ in 3D space, false positive predictions generated by different agents typically lie along their respective observation rays that intersect at $obj_1$. This property allows effective filtering of multiple false positives related to $obj_1$.
>
> 2) Suppose agent $i$ observes another real instance $obj_2$ along the observation ray of $obj_1$. However, the observation rays of $obj_2$ from other agents do not overlap with agent $i$’s observation ray of $obj_1$. Typically, there are two distinct intersection points corresponding to $obj_1$ and $obj_2$, so the true object $obj_2$ will not be mistakenly filtered out.
>
> To support this, we conduct additional targeted evaluations on crowded scenes from the OPV2V dataset:
>
> 1) We select 467 test frames in which more than 15 objects appear within the detection range. The AP70 scores with and without ROE are 74.32 and 71.24, respectively.
>
> 2) We further select 50 congested frames with over 20 objects within the detection range. In this setting, the AP70 scores with and without ROE are 66.98 and 61.36, respectively.
>
> These results demonstrate that RayFusion can effectively distinguish true positives from spurious detections even in highly crowded scenarios, confirming the robustness of the proposed occupancy-based encoding.
>
>
> $\textbf{W2: } $
>
> RayFusion performs spatial-temporal alignment on instance information from multiple agents, aiming to obtain spatial-temporal aligned and accurate instances. However, due to the inherent ambiguity of monocular depth estimation, a single agent may suffer from false positives and radial localization errors. To address this, we propose ROE, which leverages cross-view validation to effectively mitigate depth ambiguity, reduce false positive predictions, and enhance the ego’s perception performance.
> In addition, localization noise in real-world scenarios can significantly degrade the performance of collaborative systems. To tackle this issue, we introduce MIFA, which employs multi-window attention to capture both local and global spatial features in parallel. As shown in Figure 3(a), RayFusion exhibits strong robustness to localization noise.
>
>
> $\textbf{W3, Q1: } $
>
> Thank you for the valuable suggestion. A comprehensive evaluation of collaborative system robustness is crucial for deployment in real-world scenarios. Benefiting from instance features enriched with image semantics, MIFA uses multi-window attention to compensate for localization noise. In Figure 3, we analyze RayFusion’s detection performance under localization noise and communication delays, demonstrating its strong robustness against these factors.
> However, if agent $i$ predicts an instance $obj_3$ that cannot be seen by all other collaborators or the ego vehicle (e.g., due to severe occlusion), the model theoretically lacks the ability to identify or filter this instance. When $obj_3$ is visible to other collaborators or the ego vehicle but no corresponding prediction exists, we expect the model to learn to filter out such false predictions.
> Here, we further analyze the impact of corrupted poses and false positive predictions on perception accuracy:
>
> 1) Corrupted poses: We simulate corrupted poses with localization noise drawn from a Gaussian distribution with mean 0 and variance 10 m, and heading noise with mean 0 and variance 10°. Under these conditions, RayFusion’s AP70 drops from 74.25 to 33.44 (–54.96\%), while IFTR’s AP70 drops from 66.04 to 8.45 (–87.20\%).
>
> 2) False positives: We simulate false positive instances by replacing the anchor boxes of the top $N_{\text{fp}}$ high-confidence predictions from collaborators with those from low-confidence predictions. These modified instances are then merged into the collaborative instances. As $N_{\text{fp}}$ increases from 0 to 10, 50, and 100, RayFusion’s performance drops from 74.25 to 68.81, 67.68, and 66.42, respectively. This demonstrates that erroneous false positives from collaborators can indeed negatively impact model performance. In future work, we plan to further enhance robustness by integrating observations from multiple agents to suppress such unreliable predictions.

---

### Decision · Program_Chairs · 2025-09-17

**Decision:**

Accept (poster)

**Comment:**

This paper proposes RayFusion, a collaborative camera-only perception framework that introduces ray occupancy encoding to mitigate depth ambiguity in monocular 3D detection. The approach is conceptually novel, cleanly designed, and shows consistent performance gains across OPV2V, DAIR-V2X, and related benchmarks. Reviewers highlighted that the method is well-motivated (4JMZ, E2kt) and empirically validated, with robustness demonstrated under localization noise and communication delays. Reviewer kx1e raised concerns about vulnerability to faulty collaborator inputs, but additional rebuttal experiments in crowded and noisy settings partially alleviated this concern. Reviewer sx6F noted limitations in dataset scope and baseline fairness, yet acknowledged the promise of the approach. While reliance on LiDAR supervision during training slightly blunts the “camera-only” claim, the inference pipeline remains camera-based and practically relevant. This paper has received mixed scores: 2,3,5,5. The AC has gone through all the discussions in detail, and overall, the introduction of ray occupancy fusion represents a meaningful advance for collaborative visual perception. The AC finally recommends acceptance.